# Nanomaterials-Based Combinatorial Therapy as a Strategy to Combat Antibiotic Resistance

**DOI:** 10.3390/antibiotics11060794

**Published:** 2022-06-12

**Authors:** Angel León-Buitimea, Cesar R. Garza-Cárdenas, María Fernanda Román-García, César Agustín Ramírez-Díaz, Martha Ulloa-Ramírez, José Rubén Morones-Ramírez

**Affiliations:** 1Facultad de Ciencias Químicas, Universidad Autónoma de Nuevo León (UANL), San Nicolás de los Garza 66455, Mexico; angel.deb@uanl.edu.mx (A.L.-B.); cesar.r.garza.cardenas@gmail.com (C.R.G.-C.); 2Centro de Investigación en Biotecnología y Nanotecnología, Facultad de Ciencias Químicas, Universidad Autónoma de Nuevo León, Parque de Investigación e Innovación Tecnológica, Apodaca 66628, Mexico; 3Centro de Estudios Científicos y Tecnológicos No. 9 “Juan de Dios Bátiz”, Instituto Politécnico Nacional, Miguel Hidalgo, Ciudad de México 11400, Mexico; mromang1700@alumno.ipn.mx; 4Facultad de Ciencias Biológicas, Benemérita Universidad Autónoma de Puebla (BUAP), 4 Sur 104 Centro Histórico, Puebla 72000, Mexico; vampadiaz@gmail.com; 5Centro Universitario de Tonalá, Universidad de Guadalajara (UDG), Guadalajara 45425, Mexico; martha.ulloa6947@alumnos.udg.mx

**Keywords:** antimicrobial resistance, multidrug-resistant bacteria, polymers, inorganic nanoparticles, antimicrobial peptides, combinatorial treatment

## Abstract

Since the discovery of antibiotics, humanity has been able to cope with the battle against bacterial infections. However, the inappropriate use of antibiotics, the lack of innovation in therapeutic agents, and other factors have allowed the emergence of new bacterial strains resistant to multiple antibiotic treatments, causing a crisis in the health sector. Furthermore, the World Health Organization has listed a series of pathogens (ESKAPE group) that have acquired new and varied resistance to different antibiotics families. Therefore, the scientific community has prioritized designing and developing novel treatments to combat these ESKAPE pathogens and other emergent multidrug-resistant bacteria. One of the solutions is the use of combinatorial therapies. Combinatorial therapies seek to enhance the effects of individual treatments at lower doses, bringing the advantage of being, in most cases, much less harmful to patients. Among the new developments in combinatorial therapies, nanomaterials have gained significant interest. Some of the most promising nanotherapeutics include polymers, inorganic nanoparticles, and antimicrobial peptides due to their bactericidal and nanocarrier properties. Therefore, this review focuses on discussing the state-of-the-art of the most significant advances and concludes with a perspective on the future developments of nanotherapeutic combinatorial treatments that target bacterial infections.

## 1. Introduction

Since the discovery of penicillin in 1928, researchers started a promising era of antibiotic discovery and development against human pathogens; this period was referred to as the golden age of antibiotics [1,2]. During that period, medicinal chemistry was a successful strategy to produce semi-synthetic antibiotics and improve their physicochemical and pharmacokinetic properties [3]. Unfortunately, the irrational use of antibiotics, inappropriate prescriptions, extensive application in agriculture, the lack of innovation, and obstacles in regulatory approvals, have favored the emergence of antibiotic-resistant bacteria [4,5]. The resistance mechanisms developed have favored the spread of multidrug-resistant (MDR), extensively drug resistant (XDR), and pan-drug resistant (PDR) bacteria, being the latter, the most dangerous because they possess resistance to most kinds of antibiotics [6]. The most common resistance mechanisms are the enzymatic modification of the antibiotic, alterations in membrane permeability, such as efflux transporters and porins, and modification of the binding site [1,7,8]. Bacteria produce enzymes that irreparably modify antibiotics; this mechanism (enzymatic modification) is one of the most important to combat. Some of these enzymes are β-lactamases, nucleotidyltransferases, and acetyltransferases [7,8]. Another mechanism bacteria use is to reduce the internal accumulation of drugs through membrane alterations, such as the development of porins or efflux pumps. Porins are proteins found in the membrane that act as channels that allow the passive diffusion of hydrophilic molecules in Gram-negative bacteria. A modification in their expression can confer resistance to different antibiotics [7,9,10]. Efflux pumps push the antibiotic out of the cell at a high rate, which leads to conditions where drug concentrations are never high enough to cause an antibacterial effect [11]. Finally, some bacteria can develop antibiotic resistance by changing the antibiotic target site [9,10]. One of the most important and well-known examples is the methicillin resistance present in some strains of *S. aureus*. This resistance is given by the *mecA* gene, which consists of creating low-affinity binding proteins to penicillins (PBP 2a). These modified proteins prevent the main antibiotic effect of methicillin, which is to inhibit the synthesis of peptidoglycans for the cell wall; moreover, it confers immunity to other families of antibiotics such as penicillins, cephalosporins, carbapenems, and other β-lactamics [9,12]. In 2017, the World Health Organization (WHO) published a list of high-priority pathogens highlighting *Enterococcus faecium*, *Staphylococcus aureus*, *Klebsiella pneumoniae*, *Acinetobacter baumannii*, *Pseudomonas aeruginosa*, and *Enterobacter* spp., (ESKAPE). Therefore, it is urgent to find new therapies against these bacteria since they have evolved to escape the effects of different treatments and transfer their resistance to other organisms [1,9,13]. The need to treat infections, especially those caused by multidrug-resistant bacteria, implies new antibiotics development [5]. However, bacteria invariably possess the ability to evolve and develop resistance to antibiotics soon after these treatments are introduced to the clinic [14].

Various nanomaterials such as polymers (natural and synthetic), inorganic nanoparticles, and antimicrobial peptides offer an innovative platform to develop new treatment strategies [15]. In addition to their scalability, low costs, and versatility, these nanomaterials offer advantages such as lower toxicity due to lower doses, lower resistance development, and an increased antibacterial effect due to the conjunction of individual mechanisms of action [16]. One of the strategies to prevent or delay the emergence of resistance is the design of combinatorial treatments, which in many cases, have been proven to increase the antibacterial effect of the individual therapeutics [13]. Moreover, combining two or more antimicrobial agents seems to be a reasonable alternative since they may act either by inhibiting multiple targets in different pathways, inhibiting other targets in the same pathway, or inhibiting the same target in different modalities [17]. 

Therefore, this review summarizes, discusses, and concludes on the use and applications of nanomaterial-based combinatorial therapies as novel therapeutic strategies to combat infections caused by antibiotic-resistant bacteria.

## 2. New Therapeutic Strategies: Combinatorial Treatments

The discovery and preclinical development of novel antibacterial agents must involve new approaches that allow an effective and sustainable combat against antibacterial resistance [18]. There is a strong trend toward the rational design of drug combinations using direct targeting of small molecules and finding new targets or new mechanisms of antimicrobial action [19]. Different antimicrobial agents, such as polymers (synthetic and natural) [20], inorganic nanoparticles [21], and antimicrobial peptides [22], are being developed to prevent, reduce or reverse antimicrobial resistance. Therefore, we provide insights into how polymers, metal nanoparticles, and antimicrobial peptides can be applied with other antimicrobial agents as combinatorial therapies to combat antibiotic-resistant bacteria (Figure 1).

### 2.1. General Mechanisms of Antibiotic Action of Nanomaterials

Combinatorial treatments, a combination of two or more therapeutic agents, are a potential strategy to address antibiotic resistance. The application of this strategy results in a reduction in the drug dose, lower toxicity, and a decrease in the development of bacterial resistance [23]. Polymers, inorganic nanoparticles, and antimicrobial peptides, which have well-known antimicrobial properties, are used as therapeutic agents to treat bacterial infectious diseases [20,24,25,26]. Various pathogenic bacteria have been treated with combinatorial treatments. The combined effects (synergy, potentiation, or additivity) of these nanomaterials, either among themselves or with conventional antibiotics, have been reported by multiple authors [27]. In many cases, the increased antibacterial effect is due to the combination of the individual drug effects [28].

For example, AMPy enhances the therapeutic effectiveness of antibiotics due to their well-defined globular structure and their ability to attach drugs onto their surface. They also reduce β-lactamase enzyme activity and disrupt cell walls. AMPy can complex antibiotics (bioconjugates) by forming a stable ion-pairing. Thus, they promote damage to the cell walls allowing the release of complexed antibiotics [29]. Moreover, AMPy can serve as a polymeric drug delivery system. They enhance the safety and efficacy of the other component by regulating the rate, time, and place of release in the body [30]. Finally, AMPy is utilized as a capping agent capable of inducing subtle changes in the nanomaterial, enhancing its therapeutic effects [31].

Regarding the inorganic nanoparticles, there are several possible modes of action on the bacteria (Figure 2): (a) alteration of the bacterial cell wall and membrane, (b) induction of oxidative stress due to excessive intracellular production of reactive oxygen species (ROS), (c) inhibition of crucial proteins/enzymes and DNA, (d) disruption of metabolic pathways, and (e) intracellular accumulation of metal ions released from INP [32]. Therefore, these nanomaterials have been proved to be most effective against different strains of pathogenic bacteria.

Antimicrobial peptides are cationic (positively charged) and amphiphilic (hydrophilic and hydrophobic) molecules, and these properties are related to their ability to interact with the bacterial cell membranes [33,34]. In general, AMPs may act through two main mechanisms; in the first one, the AMPs induce membrane disruption and cell lysis, causing bacterial death. The second one is when the AMPs enter cells (without membrane disruption) and interfere with an intracellular pathway or bind to nucleic acids. Thus, AMPs are potential candidates for developing novel antimicrobial therapeutics.

### 2.2. Nanomaterials-Based Combinatorial Treatments

#### 2.2.1. Polymers

Thanks to advances in chemistry research, there is the possibility of designing and synthesizing antimicrobial polymers (AMPy), materials capable of inhibiting or killing bacteria [20]. AMPy can display antibacterial activities through their chemical structures, such as chitosan compounds with quaternary nitrogen groups, halamines, and poly-ε-lysine (ε-PL).

In this review, we summarize several contributions of combinatorial treatments that include synthetic and natural polymers as the main component of effective antimicrobial agents and their effectivity on bacteria. Table 1 shows the combinatorial treatments that incorporate synthetic and natural polymers as the main component of effective antimicrobial agents.

##### Synthetic

Although several investigations have demonstrated the antimicrobial effect of individual synthetic polymers, only a few reports include their combination with antibiotics or other molecules as combinatorial treatments [35]. Some of the most critical literature include:

##### Poly (Lactide-Co-Glycolide)

Poly (lactic-co-glycolic acid) (PLGA) is a synthetic functional polymer commonly used in the biomedical field due to its advantages, including biodegradability, biocompatibility, and non-toxicity [36]. Besides being one of the polymers approved by the FDA for use in humans, PLGA is widely used in drug delivery systems such as microspheres and nanoparticles, proven to be very efficient [37]. Several studies have shown that grafting an AMP (magainin II) by covalent immobilization to an electrospinning PLGA membrane reduced the number of adhered bacteria (*E. coli* and *S. aureus*) by more than 50% [38]. Another strategy that has sparked the interest of several authors is the incorporation of silver nanoparticles (AgNPs) into polymeric matrices. Authors showed that the integration of AgNPs within the PLGA matrix was carried out by creating nanofiber scaffolds, which inhibited bacterial growth (*P. aeruginosa, K. pneumoniae*, *S. saprophyticus*, and *E. coli*) depending on the concentration of AgNPs incorporated within the scaffold [39]. Furthermore, Zhu et al. developed an anti-biofilm system, drug-free, using cationic nanoparticles (CNPs). The bioactivity of the CNPs against *Streptococcus mutans* was examined, and the results showed a concentration-dependent activity against bacteria, an excellent inhibition in biofilm formation, and an interruption of mature biofilms [40].

Unfortunately, some studies showed cytotoxic effects on several cell lines due to the presence of AgNPs in polymeric matrices [41]. For example, Mohiti-Asli et al. showed that human epidermal keratinocytes and human dermal fibroblasts’ viability were decreased by an increase in silver concentration within the coating solution [42]. Moreover, some authors have explored the coupling of antibiotics to polymeric matrices to improve drug delivery [43]; however, not all combinations were successful. For example, roxithromycin (ROX) delivery was enhanced by anchoring to cyclodextrins (ROX-CD) and encapsulating them in PLGA NPs (ROX-CD/PLGA). Results showed that ROX/PLGA NPs dual combination was more potent in inhibiting the growth of selected MDR bacterial strains than the ROX-CD/PLG triple combination. This difference was basically due to the formation of the strong electrostatic interactions among the individual components, which interfere with the release of the drug from these formulations [44].

One strategy to incorporate antimicrobial properties into biomaterials such as PLGA is described by Qian et al. They showed that the PLGA/PCL (polycaprolactone) electrospun scaffold coated with collagen, and modified with silver, had enhanced biocompatibility, osteogenic, and antibacterial properties (against *S. aureus* and *Streptococcus mutans*) [45]. Another exciting strategy involved synthesizing organic-metallic hybrid systems, which contain AgNPs integrated into biodegradable polymer nanofibers such as PLGA [20]. These systems coupled benefits from both individual components, including a greater reaction surface area and higher permeability from PLGA with the antimicrobial properties of AgNPs [46]. Similarly, TiO_2_ nanoparticles have also been incorporated into polymeric systems. PLGA is one of the main polymers used in different techniques, and its antibacterial properties against a variety of microorganisms have been detected [47]. These studies evaluated the viability of the biofilms composed of PLGA-TiO_2_-NP under ultraviolet light irradiation, showing that biofilms for artificial deposit applications that contained one-tenth of TiO_2_-NP were effective against *E. coli* and *S. aureus* [47]. Despite the various applications of PLGA-incorporated TiO_2_-NPs, it has been shown that the pure PLGA solution, subjected to a moderate solvent removal process, can generate a tightly arranged thin film with fewer defects than when it is mixed with TiO_2_-NPs. The structure of pure PLGA is disrupted, leading to structural defects and increased permeability of the biofilm. Moreover, a composite biofilm with a high concentration of TiO_2_ had a correspondingly higher number of structural defects and is more fragile than composite biofilms with a low concentration of TiO_2_.

**Table 1 antibiotics-11-00794-t001:** Combinatorial treatments that include synthetic and natural polymers as the main component of effective antimicrobial agents.

Nanomaterial	Combined with (Rate/Ratio)	Form	Size	Targeted Bacteria	Antimicrobial Effects	References
Synthetic						
Poly (lactide-co-glycolide)	Magainin II (0.2 ± 0.05 μg/cm^2^)	PLGA nanofibers Magainin II covalently immobilized	PLGA nanofibers diameter 715 ± 45 nm	*Escherichia coli* *Staphylococcus aureus*	Reducing the number of adhered bacteria	[38]
	AgNPs (3% *wt*/*v*)	Nanofibers of PLGA AgNPs within the scaffold	Nanofiber diameters between 487 and 781 nm AgNPs diameter < 100 nm	*P. aeruginosa, K. pneumoniae, S. saprophyticus,* and *E. coli*	Inhibition of bacterial growth	[39]
	Poloxamer 188 (0.1% *w*/*v*)	Nanospheres of PLGA Poloxamer 188 coating	Nanospheres diameter 217.7 nm	*Streptococcus mutans*	Inhibition of planktonic bacterial growth and biofilm formation, and disrupted ∼70% mature biofilm	[40]
	Polycaprolactone (PCL, nd) Type I collagen (2% *w*/*v*) AgNPs (nd)	Nanofibers of PLGA/PCL AgNPs reduced in situ with nanofibers Collagen coating	Nanofibers diameter 477 ± 186 nm,	*S. aureus* and *Streptococcus mutans*	Antibacterial properties	[45]
	TiO_2_ NPs (10% *w*/*w*)	TiO_2_/PLGA composite biofilms	TiO_2_ NPs diameter 20 nm	*E. coli* and *S. aureus*	Antibacterial properties	[47]
Poly (glycolic acid)	ε-caprolactone (14%) trimethylene carbonate (14%) Oxygen plasma treated	Monofilament suture	nd	*E. coli K12*	Antibacterial properties	[48]
	N-halamines polymers	PGA sutures N-halamines coating via layer-by-layer	nd	*S. aureus* and *E. coli*	Effective bactericide properties	[49]
	PLGA (30:70 PGA/PLGA) AgNPs (3% wt)	PGA: PLGA fibers AgNPs within the scaffold	Nanofibers diameter 1170 ± 166.98 nm AgNPs diameter 22 nm	*S. aureus* and *E. coli*	Antibacterial activity	[50]
	PLGA (50:50 PLGA/PGA) AuAg core/shell NPs (600 mg/kg of stent)	PGA/PLGA ureteral stent AuAg core/shell nanospheres	Au core diameter 10.94 nm Ag shell thickness 6.98 nm	*E. coli* and *S. aureus*	Long-lasting inhibitory activity and remarkable antibiofilm properties.	[51]
	Propylene fumarate (PPF, co-polymer) Graphene oxide (GO, 5% wt) Hydroxyapatite (HA, 20% wt).	PGA/PPF nanofibers HA nanorods and GO within the scaffold	Nanofibers diameter 469 nm HA nanorods diameter 18 nm and length 50–80 nm	*S. aureus* and *E. coli*	Extensive biocidal activity.	[52]
**Natural**						
Chitosan	Multiwalled carbon nanotubes (5 × 10^−3^% wt)	Chitosan/MWCNT biocomposites	-	ESKAPE group bacteria	Improved antimicrobial activity	[53]
	Chlorhexidine (3% *v*/*v*)	Chitosan nanoparticles Chlorhexidine functionalization	Chitosan nanoparticles diameter 70.6 ± 14.8 nm	*Enterococcus faecalis*	Improved antibacterial activity	[54]
	Nisin (0.625 g/L) Tea polyphenols (0.313 g/L)	Chitosan, Nisin and Tea polyphenols in dissolution	-	Gram-negative and Gram-positive bacteria	Improved antimicrobial activity	[55]
	Zinc-EDTA chelate	Chitosan solution Zinc-EDTA chelate solution	-	*Penicillium italicum*	Better inhibitory activity	[56]
Inulin	Modified (amphiphilic amino inulin)	Chemical modification of inulin	Amphiphilic amino inulin in solution	*S. aureus*	Antibacterial activity	[57]
	Chitosan (1% *w*/*v*)	Inulin was glycated to chitosan in solution	-	*Staphylococcus aureus* ATCC 25923, *Escherichia coli* ATCC 25922, *Pseudomonas aeruginosa* ATCC 27853, *Bacillus subtilis* ATCC 23857, *Candida albicans* PTCC 5027 and *Aspergillus niger* ATCC 23857	Significant antimicrobial activity	[58]
	Chitosan (nd)	Covalent conjugation of inulin to chitosan in solution	-	*S. aureus*	Significantly improved	[59]
	Polyvinyl alcohol (PVA) (15% *w*/*v*)	Composite nanofibers of crosslinked Inulin and PVA	Nanofiber diameter widely dispersed	*E. coli* and *S. aureus*	Increased antibacterial activity	[60]
	Carboxymethylcellulose (CMC films) Celullose nanofiber (CNF, 2.5%) *L. plantarum* (10^9^ CFU/mL)	CMC films incorporated with inulin, CNF *L. plantarum* inoculated on the film	CMC films CNF diameter 35 nm, length 5 μm	*S. aureus*, *E. coli,* and *K. pneumoniae*	Antibacterial activity	[61]
Alginate	ZnO NPs (nd) Cellulose fibers	Cellulose cotton fibers impregnated with sodium alginate-ZnO NPs ZnO NPs “rod-shape”	ZnO NPs diameter 25 ± 5 nm	*E. coli*	Significant antibacterial activity	[62]
	Copper (Cu, ~100 µmol/g of microbed)	Cu-alginate spherical microbeds	Cu-alginate microbeds diameter ~550 μm	*E. coli* and *S. aureus*	Bactericidal effects	[63]
	Hydroxyapatite nanoparticles (HA NPs, 5% *w*/*w*)	Alginate-HA NPs nanocomposite film -Spherical HA-NPs	Alginate-HA NPs film thickness 0.036 ±0.002 mm HA NPs diameter 25 ± 2 mm	*Listeria monocytogenes*	Showed the highest antibacterial effect	[64]
	AgNPs (nd)	Alginate-AgNPs solution Spherical AgNPs	AgNPs diameter < 50 nm	*S. aureus* and *E. coli*	Increased membrane permeability and disruption of the bacterial wall	[65]
	Graphene oxide (GO, 1% *w*/*w* alginate) Zinc (Zn, 12% *w*/*w* alginate)	Alginate-GO cross-linked films Zn covering	-	*S. aureus*	High antibacterial activity	[66]
	Hydroxypropyl methylcellulose (HPMC, 1% *w*/*w*) ε-polylysine (ε-PL, 1% *w/w*)	Alginate-HPMC-ε-PL films	Film thickness 18 ± 6 µm	*E. coli* and *S. aureus*	99.9% bacterial reduction	[67]
	Corona treated Polypropylene (CPP, nd) Copper oxide nanoparticles (CuO NPs, nd)	CPP-alginate fiber nanocomposite CuO NPs reduced in matrix	CuO NPs diameter 43 ± 15 nm	*E. coli*, *S. aureus*, and *Candida albicans*	Excellent antimicrobial activity	[68]

nd: not described.

Poly (Glycolic Acid)

Poly (glycolic acid) (PGA) is a biopolymer used for biomedical applications due to its biodegradability and thermal and mechanical properties [69]. Another notable feature of PGA is its easy de-esterification to monomer units, leading to an enhanced metabolization and, therefore, a faster degradation rate [70].

Chemical modifications perpetrated on the surface of a polymeric material can significantly control its antimicrobial properties. An example can be drawn from the nano structuration induced by oxygen plasma treatment on modified PGA (PGA (72%), ε-caprolactone (14%), and trimethylene carbonate (14%) absorbable monofilament sutures; this treatment led to materials with antimicrobial properties against *E. coli* K12 [48]. Likewise, an N-halamine-modified PGA multifilament obtained by the layer-by-layer technique produced effective sutures that successfully inactivated *S. aureus* and *E. coli* strains within the first 30 min of contact [49]. In addition, PGA-PLGA electrospun nanofiber sutures added with 3% AgNPs have shown antibacterial activity against Gram-positive (*S. aureus*) and Gram-negative (*E. coli*) bacteria. On the other hand, the results of the in vitro degradation test showed that the mechanical properties of these sutures decreased rapidly due to the presence of AgNPs. [50]. Moreover, PGA-PLGA membranes with embedded gold and silver nanoparticles have shown remarkable antibiofilm properties and long-lasting inhibitory properties of 99%, with removal times of between 5 and 10 min for *E. coli* and *S. aureus* [51]. Lately, electrospun fibers of a novel biodegradable PGA and propylene fumarate (PGA-co-PPF) copolymer with graphene oxide (GO) and hydroxyapatite nanorods (HA) have been reported to display extensive biocidal activity against Gram-positive (*S. aureus*) and Gram-negative (*E. coli*) bacteria [52].

Natural

Also known as biopolymers, natural polymers are materials biologically derived from living organisms and are primarily studied due to their biodegradable, bioactive, or in some cases, antimicrobial properties [71,72]. An essential characteristic of natural polymers is that they can be modified to design their properties, turning them into semi-synthetic polymers with active functional groups [71,73]. Additionally, they are affordable, environmentally friendly, and less expensive than synthetic polymers [73].

Chitosan

Chitosan, derived from the alkaline deacetylation of chitin, is a linear high molecular weight compound consisting of two monosaccharides, N-acetyl-D-glucosamine and D-glucosamine, linked by glucosidic β-1-4 bonds [74]. Among the most important biological properties, its biodegradability through hydrophilic enzymes stands out, in addition to its bactericidal activity [75]. Chitosan’s antimicrobial activity is attributed to the presence of amino groups in its composition, allowing it to interrupt the normal functions of the bacterial membrane, causing both the leakage of intracellular components and the breakdown of nutrient transport to the cell [76]. However, a disadvantage of chitosan is that its antibacterial activity occurs at pH < 6, so it must be synthetically modified [77] to study it under neutral and physiological conditions.

The combination of chitosan derivatives with other compounds has exhibited significant bactericidal activity. For example, chitosan-based biocomposites with multi-walled carbon nanotubes (MWCNT) are effective antimicrobial agents against some bacteria from the ESKAPE group [53]. Furthermore, the antibacterial activity of chitosan against *Enterococcus faecalis* has been significantly improved by combining it with chlorhexidine in endodontic sealants [54]. Moreover, two effective combinatorial strategies against resistant bacteria have been studied; one determined that the optimal combination of nisin, tea polyphenols (TP), and chitosan (0.625, 0.313, and 3.752 g/L, respectively) has inhibitory properties against Gram-negative and Gram-positive bacteria [55]; the second, demonstrated the synergy of chitosan hydrolysates combined with divalent metal ion-EDTA compounds. Finally, active molecular chitosan (AMC) was found to have better inhibitory activity against *Penicillium italicum* in the presence of the zinc-EDTA chelate compared to other combinations [56].

Inulin

Inulin is a natural polysaccharide and oligomer widely used in the food industry and generally composed of between 12 and 15 units of fructose [78]. Besides its fructose units, it usually contains a reducing end of glucopyranose units (GFn). Inulin is mainly extracted from low-requirement cultures such as *Helianthus tuberosus*, chicory, and yacon [79]. Chemical modifications are often introduced to enhance the properties of inulin before use. For example, the “click chemistry” method has been used to modify inulin at its primary hydroxyl groups, leading to the synthesis of amphiphilic amino inulin. Amphiphilic amino inulin has exhibited antibacterial activity against *S. aureus*, with an inhibition index of 58% at 1 mg/mL [57].

An interesting combination is coupling inulin with probiotics *Lactobacillus* sp. and *Lactococcus* sp. These systems have been proven symbiotic with regulatory properties for the growth of beneficial bacteria and antibacterial activity [80]. On the other hand, chitosan-inulin conjugates were obtained through the Maillard reaction with different pHs, corroborating that the conjugates with low pH values presented a more significant antimicrobial activity [58]. Another application of the inulin–chitosan conjugates is against bacterial biofilms. It has been shown that they significantly improved the activities against *S. aureus* biofilms and greatly inhibited their formation [59]. Wahbi et al. demonstrated that composite nanofibers (CNF) of inulin/polyvinyl alcohol (PVA) had an increased antibacterial activity against *E. coli* and *S. aureus* [60]. Finally, a probiotic nanocomposite film based on carboxymethylcellulose (containing cellulose nanofiber) and inulin was developed. It displayed antibacterial activity against nine pathogens, including *S. aureus*, *E. coli,* and *K. pneumoniae* [61].

Alginate

Alginate is a natural polysaccharide composed of mannuronic acid and guluronic acid. It has remarkable biocompatibility and biodegradability properties and is non-toxic to human cells [81]. This biopolymer is extracted mainly from brown algae (*Phaeophyceae*) through treatments with aqueous alkaline solutions. The extract is filtered and mixed with calcium chloride to obtain the alginate salt precipitate. The alginate salt is then converted to alginic acid by treating it with dilute hydrochloric acid (HCL) to purify it next to produce the water-soluble sodium alginate (Na-Alg or SA) [81].

Recently, the antimicrobial properties of sodium alginate (SA) have been studied in combination with other compounds; an example of this is zinc oxide-cellulose sodium alginate nanocomposite fibers (ZnO-SACNF) that have shown significant antibacterial activity against *E. coli* [62]. At the same time, copper alginate hydrogels in the form of microspheres produced immediate bactericidal effects against *E. coli* and *S. aureus* [63]. Moreover, it has been found that sodium alginate film with 5% hydroxyapatite nanoparticles (HA NPs) showed the highest antibacterial effect against *Listeria monocytogenes* [64]. Another study described the green synthesis of sodium alginate (SA)-silver nanoparticles (AgNPs), where SA acted as a stabilizer for AgNPs. These NPs induced cell death due to increased membrane permeability and disruption of the bacterial wall in *S. aureus* and *E. coli* [65].

Sodium alginate has also been used to synthesize zinc alginate films with and without 1% graphene oxide (GO), showing high antibacterial activity against *S. aureus*. Unfortunately, several mechanical properties were affected by incorporating 1% *w*/*w* of GO [66]. Similarly, crosslinked alginate/hydroxypropyl methylcellulose/ε-polylysine films with low content of plasticizers showed 99.9% bacterial reduction in both *E. coli* and *S. aureus* strains. It is noted that ε-PL should be used moderately because it competes for alginate chains, making the films more fragile [67]. Lastly, a novel antimicrobial nanocomposite was developed based on corona treated polypropylene (CPP), added with alginate and copper oxide nanoparticles; this nanocomposite exhibited excellent antimicrobial activity against *E. coli*, *S. aureus*, and *Candida albicans* [68].

##### Electrospinning and 3D Printing as a Novel Polymer Synthesis Technology

Among the most recent synthesis strategies and technologies to produce nanomaterials and their combinations, electrospinning and three-dimensional (3D) printing holds great promise as a production method for scaffolds, nanofibers, and supporting nanomaterials [82]. Electrospinning is a simple and versatile method for creating nanofiber materials and scaffolds with various structures and surface areas, which has a broad application field. Electrospinning involves converting a liquid polymer solution into solid nanofibers by applying an electrical force in the presence of a strong electric field [77]. Interestingly, electrospun nanofibers materials based on synthetic and natural antibacterial polymers have been studied due to their biocompatibility, biodegradability, and non-toxicity [83,84]. On the other hand, 3D printing is a relatively new, rapidly expanding method of manufacturing that allows the formation of 3D compact structures with desired and predefined architecture [85]. The use of nanomaterials (including a wide range of polymers, metals, composites, or natural products) in 3D printing is gaining attention due to the tremendous functionality this approach provides for those materials [86,87]. Nevertheless, further research is needed in this area, including using other polymeric materials, antimicrobial agents, and methods of producing filaments with selected properties.

#### 2.2.2. Inorganic Nanoparticles

Another alternative to combat the ESKAPE bacteria is inorganic nanoparticles (INPs). Some of the most important INPs are gold nanoparticles (AuNPs), silver nanoparticles (AgNPs), magnetite NPs (MNPs), titanium oxide nanoparticles (TiO_2_NPs), zinc oxide nanoparticles (ZnONPs), and Ag-Au core-shell NPs [88,89,90]. INPs inhibit bacterial growth through different mechanisms [91,92]. However, combinatorial treatments of INPs with antibiotics, polymers, and antimicrobial peptides, can produce a synergistic effect and reduce the therapeutic doses [93,94].

The combination between INPs and polymers can be achieved through two strategies: using the polymer as a coating for the INPs or incorporating the nanoparticles into the polymer matrix [95,96]. There are different approaches to coating the surface of a nanoparticle with polymers; the most recognized is the modification of the surface through chemical treatments, ligand exchange techniques, and grafting techniques. Chemical surface treatments refer to a method of changing the structure and state of the surface of a nanoparticle by chemical reaction or chemisorption between the surface of the nanoparticle and the treatment agent [97,98]. The ligand exchange technique involves the substitution of one or more ligands in a complex ion with one or more different ligands. Usually, in nanoparticles, this technique is used to change the capping agent used in the original synthesis [97,99]. Grafting polymers to the surface of a nanoparticle enhances the chemical functionality and alters the surface of the INPs. Because monomers usually have a low molecular weight, they can penetrate nanoparticles and react with the activated sites on the nanoparticle surface, becoming partially filled with grafted macromolecular chains. Therefore, the aggregated nanoparticles become further separated [100,101].

On the other hand, two general approaches can be used to incorporate INPs into polymers: in situ and ex situ. In situ refers to using the polymer matrix as the reaction medium. Typically, the polymer is chemically modified to incorporate functional groups into its matrix that serve to reduce and form INPs on itself. Contrary to the in situ methodology, ex situ refers to the fact that the INPs particle is synthesized before incorporating into the polymer [95]. There is a recent interest on the part of the scientific community to create nanoscale polymeric structures to use them as nanocarriers. Some research groups use these nanovehicles to improve the delivery or transport of new therapies such as antimicrobial peptides [102]. Meanwhile, green synthesis has also led different research groups to create new synthesis routes for combinatorial therapies, facilitating the elimination of toxic compounds involved in conventional chemical synthesis and unwanted by-products by taking advantage of different organic residues [103,104].

Table 2 summarizes the combinatorial treatments that include inorganic nanoparticles as the main component of effective antimicrobial agents.

##### Gold Nanoparticles

Gold nanoparticles (AuNPs) are of great interest due to their physicochemical, optical, biocompatibility, and low toxicity properties. Several authors have evaluated the antimicrobial activity of AuNPs; however, there is still no consensus on whether this nanomaterial has antimicrobial activity. Although several authors have shown that AuNPs do not have antibacterial activity [105,106,107,108,109,110,111,112,113], others have shown that they do [114,115,116,117]. Despite this, in this review, we analyzed the combinatorial therapies where AuNPs were functionalized with different protection agents and polymers or served as a nanocarrier for antibiotics and antimicrobial peptides, among others.

For example, Feng et al. evaluated the activity of AuNPs functionalized with N-heterocyclic compounds and assessed the antimicrobial effect against methicillin-resistant *S. aureus* (MRSA) and MDR *P. aeruginosa*. The results showed that 2-mercaptoimidazole-functionalized AuNPs had low cytotoxic activity in HUVEC cells (human umbilical vein endothelial cell) and excellent antibacterial effect in both strains [118]. Besides, Sun et al. synthesized AuNPs with 4,5-diamino-2 pyrimidiethiol (DAPT) and bovine serum albumin (BSA). This compound (AuNPs- DAPT-BSA) exhibited a bactericidal effect (killed up to 99%) in MDR *E. coli* and MRSA bacteria [119].

In recent years, the scientific community has focused on producing eco-friendly nanoparticles using compounds from extracts of plants, fungi, bacteria, algae, and actinomycetes, to reduce costs and mitigate the toxic effects of their precursors [120,121,122,123,124,125]. A study demonstrated that AuNPs synthesized from the extract of *Garcinia mangostana* L. combined with conventional antibiotics possess a synergistic antimicrobial effect against clinical isolates of *Staphylococcus* spp. and *Pseudomonas* spp. The results showed that AuNPs combined with azithromycin and streptomycin increased the antibacterial activity (34.8% and 33.3%, respectively) against *Staphylococcus* spp. compared to antibiotics alone. On the other hand, the combination of AuNPs with penicillin and azithromycin exhibited better antimicrobial activity (75% and 50%, respectively) against *Pseudomonas* spp. [126].

Combining AuNPs with polymers is another promising treatment strategy against resistant bacteria [127,128,129]. For example, Pradeepa et al. synthesized AuNPs with an exopolysaccharide (EPS) (extracted from *Lactobacillus plantarum*) and functionalized them with antibiotics (e.g., ciprofloxacin, levofloxacin, ceftriaxone, and cefotaxime). The treatments showed a synergistic antibacterial effect against MDR strains of *K. pneumoniae*, *S. aureus,* and *E. coli*. The combinations that showed better results were AuNP-ciprofloxacin against *K. pneumoniae* and *E. coli*; and AuNP-levofloxacin against *S. aureus* [130]. Furthermore, Li et al. showed that bacterial cellulose decorated by 4,6-diamino-2-pyrimidinotiol-modified AuNPs inhibited bacterial growth of *E. coli*, MDR *E. coli*, *P. aeruginosa,* and MDR *P. aeruginosa* [131]. Finally, a bioconjugate containing AuNPs, polycobaltocenium (PCo) homopolymer, and penicillin-G (Peni) exhibited remarkable antimicrobial efficiency (synergistic effect) against *S. aureus* and *E. coli* compared to the individual PCo-Peni and Peni treatments [132].

**Table 2 antibiotics-11-00794-t002:** Combinatorial treatments that include inorganic nanoparticles as the main component of effective antimicrobial agents.

Nanomaterial	Combined with (Rate/Ratio)	Form	Size	Targeted Bacteria	Antimicrobial Effects	References
Gold Nanoparticles (AuNPs)	2-mercaptoimidazole (MI, 10:1 AuNPs)	Spherical AuNPs MI capping	AuNPs diameter ~ 3.5 nm	MRSA MDR *P. aeruginosa*	Excellent antimicrobial effects with low cytotoxic activity in HUVEC cells.	[118]
	4,5-diamino-2 pyrimidiethiol (DAPT) Bovine serum albumin (BSA) (21:1 BSA/DAPT)	Spherical AuNPs DAPT and BSA capping	AuNPs diameter 4.11 ± 0.32 nm	MDR *E. coli*	Killed up to 99% of bacteria	[119]
	Azithromycin (Azi, 3:1 Azi/AuNPs) Streptomycin (Sty 1:1 Sty/AuNPs)	Spherical AuNPs AuNPs disc impregnated with antibiotic solution	AuNPs diameter between 20 to 40 nm	Clinical isolates *Staphylococcus* spp.	Increased antibacterial activity compared to antibiotics alone	[126]
	Penicillin G (PeG, 1:5 PeG/AuNPs) Azithromycin (3:1 Azi/AuNPs)			Clinical isolates *Pseudomonas* spp.		
	Ciprofloxacin (4.3 µg of antibiotic conjugated/mL)	Spherical AuNPs Antibiotic conjugation to AuNPs surface	Bare AuNPs diameter 10–20 nm Functionalized AuNPs diameter 20–30 nm	MDR *K. pneumoniae* MDR *E. coli*	Synergistic antibacterial effect	[130]
	Levofloxacin (3.87 µg of antibiotic conjugated/mL)			MDR *S. aureus*		
	Bacterial cellulose (BC) 4,6-diamino-2- pyrimidinotiol (Au-DAPT, 3.3 ± 0.3 µg/cm^2^)	BC membrane for wound dressing decorated Spherical AuNPs capped with DAPT	Au-DAPT NPs diameter ≈3 nm	*E. coli*, MDR *E. coli*, *P. aeruginosa* and MDR *P. aeruginosa*	Inhibited bacterial growth	[131]
	Polycobaltocenium homopolymer (PCo, 38% *w/w*) Penicillin G (PeG 27% *w/w*)	Spherical AuNPs capped with PCo and functionalized with PeG	Bare AuNPs diameter 2–3 nm Functionalized AuNPs diameter to 6 nm (Au@PCo)	*S. aureus* *E. coli*	Synergistic effect compared with individual treatments	[132]
Silver nanoparticles (AgNPs)	Vancomycin, Oleandomycin, Ceftazidime, Penicillin G, Novobiocin, Carbenicillin, Lincomycin, and Erythromycin (15 µg/disc)	Spherical AgNPs Antibiotic disk impregnated with AgNPs (500 ppm)	AgNPs diameter 15–20 nm	MDR *P. aeruginosa* MDR *E. coli*	Synergistic effect compared with individual treatments	[133]
	Ampicillin (Amp) and amikacin (Amk) (1:1 Antibiotic/AgNPs)	Spherical AgNPs functionalized with antibiotics	Bare AgNPs diameter 8.57 ± 1.17 AgNPs +Amp diameter 4.01 ± 0.80 AgNPs +Amk diameter 6.03 ± 0.87	Clinical isolates of *E. faecium, S. aureus*, *A. baumannii, Enterobacter cloacae, E. coli, K. pneumoniae,* and *P. aeruginosa*.	Synergistic, partial synergistic and additive antibacterial effects among the different combinations	[134]
	Bacteriocin extracted from *Lactobacillus paracasei* (nd)	Spherical AgNPs conjugated with bacteriocin	AgNPs diameter ~16 nm	Clinical MDR isolates of *S. aureus,* *P. aeruginosa, K. pneumoniae, E. coli*, and *Staphylococcus pyogenes*	Synergistic bactericidal effect compared to individual treatments	[135]
	Polyvinyl alcohol (PVA) Chitosan (CS)	PVA-AgNPs and CS-AgNPs nanocomposite films Spherical AgNPs	AgNPs diameter ~15 nm	Clinical isolates of *S. epidermis, S. aureus, K. pneumoniae*, and *E. coli*	Remarkable antimicrobial effect and inhibition of biofilm production	[136]
Zinc Oxide nanoparticles	Cefepime (0.0256 μg/mL) Ampicillin (0.001 μg/mL)	Antibiotics in solution Spherical ZnO NPs (80 μg/mL)	ZnO NPs diameter ~15 nm	Clinical isolates of *E. coli*	Synergistic effect	[137]
(ZnO NPs)	Cephotaxime (0.032 μg/mL) Ceftriaxone (0.1 μg/mL ceftriaxone)	Antibiotics and NPs in solution Spherical ZnO NPs (60 μg/mL)		Clinical isolates of *K. pneumoniae*		
	Ciprofloxacin (8 mg/mL) Ceftazidime (32 mg/mL)	Antibiotics and NPs in solution Spherical ZnO NPS	ZnO NPs diameter ~17.08 nm	Clinical isolates of *A. baumannii*	Increased antimicrobial activity to overcome bacterial resistance	[138]
	Ciprofloxacin (nc)	Ciprofloxacin conjugated to ZnO NPs Multiple shapes of ZnO NPs	ZnO NPs diameter 20–24 nm	*Klebsiella* spp. and *E. coli*.	Increased antibacterial activity compared to individual treatments	[139]
	Colistin (1–4 μg/mL)	Colistin and ZnO NPs in solution ZnO NPs form n.d	ZnO NPs diameter 50 nm	Clinical isolates of *P. aeruginosa*	Synergistic effect	[140]
	Chitosan NPs (1:1 ZnO NPs/chitosan)	Chitosan NPs and ZnO NPs in solution ZnO NPs form n.d	ZnO NPs n.d.	MDR *E. coli* MDR *E. faecium*	Synergistic effect	[141]
	Lipid micelle (5:8 mass Lipid/ZnO NPs) Chitosan (5:24 mass chitosan/ZnNPs	Lipid nanomicelles Spherical ZnO NPs inside micelle Chitosan capping micelles ZnO NPs form n.d	Micelle diameter ~338.7 nm ZnO NPs n.d.	MDR *E. faecium*	50% reduction in bacterial biofilm formation	[142]
	EPS from *Rhodotorula mucilaginosa* UANL- 001L (2 mg/mL)	EPS-ZnO NPs nanobiocomposite ZnO NPS without defined shape	ZnO NPs diameter 8.32±1.99 nm.	MDR *P. aeruginosa* MDR *S. aureus*	Inhibition of bacterial growth (50–80%)	[143,144]
					No visible toxic effects in a Wistar rat model	
Titanium dioxide nanoparticles (TiO_2_ NPs)	Two geometric isomers ferrocene-carborane derivatives (FcSB, 0.5–1:4 FcSB/ TiO_2_ NPs)	FcSB and TiO_2_ NPs in solution Spherical TiO_2_ NPs	TiO_2_ NPs diameter 41 ± 12 nm	Clinical MDR isolates of A. *baumannii*	100% inhibition of growth	[145]
	ZnO NPs (nd)	TiO_2_ NPs and ZnO NPs in solution Spherical TiO_2_ NPs and ZnO NPs	TiO_2_ NPs and ZnO NPs diameter between 20–50 nm	Clinical MDR isolates of *A. baumannii*, and *K. pneumoniae*	Additive effects	[146]
	Silver ions (Ag^+^ 8% *w*/*w*)	TiO_2_ anathase phase NPs shell with Ag^+^ incorporated Spherical TiO_2_ NPs	TiO_2_ NPs diameter 200 ± 10 nm with a wall thickness of 20–30 nm	MDR *S. aureus*	Strong antibacterial activity	[147]
	Polytetrafluorethylene (PTFE, 2 g/L)	TiO_2_ NPs- PTFE particles coated in a stainless-steel surface	TiO_2_ NPs diameter < 25 nm, PTFE particles 200–300 nm	*E. coli* *S. aureus*	Antibacterial and anti-adhesion properties.	[148]
	ZnO NPs (1:3 TiO_2_ NPs/ZnO NPs)	TiO_2_ NPs and ZnO NPs in solution Shape n.d.	Size n.d.	*S. aureus* ATCC29213, *E. coli* ATCC 25922, MRSA ATCC 38591, and *K. pneumoniae* ATCC 700603	Bactericidal activity	[149]
				MRSA MDR *K. pneumoniae*	50% reduction in biofilm	
	Cefepime Ceftriaxone Amikacin Ciprofloxacin (Sub-MIC values)	TiO_2_ NPs and antibiotics in solution Irregulate shape TiO_2_ NPs	TiO_2_ NPs particle size 64.77 ± 0.14 nm	MDR *P. aeruginosa*	Synergistic effect	[150]
	Erythromycin (2–16 mg/L)	TiO_2_ NPs and erythromycin in solution “Round-shape” TiO_2_ NPs	TiO_2_ NPs size 15–18 nm.	MRSA	Synergistic effect	[151]
	Silver (1.4% of nanoparticle)/ rifampicin, doxycycline, ceftriaxone, and cefotaxime (66.4, 60.3, 34.0 and 23.6 μg/mg of nanocomposite, respectively)	Antibiotics attached via electrostatic interactions Fe_3_O_4_/Ag NPs Roundish shape Fe_3_O_4_/Ag NPs	Fe_3_O_4_/Ag NPs diameter 40–50 nm in size	*S. aureus* and *Bacillus pumilus*	Antibacterial properties	[152,153,154]
Magnetite nanoparticles (Fe_3_O_4_ NPs)	Cefepime (3.53 ± 0.1% *w/w* of NP PLGA (film)	PLGA/Fe_3_O_4_-Ce NPs composite films -Spherical Fe_3_O_4_ NPs functionalized with cefepime (Fe_3_O_4_-Ce NPs)	Fe_3_O_4_/Ce NPs diameter ~5 nm	*S. aureus* and *E. coli*	Suitable materials for the sterilization on implantable devices, biocompatible and efficient inhibition of bacterial biofilm	[155]
	Eugenol	Fe_3_O_4_ NPs functionalized with Eugenol “Quasi-spherical” shape Fe_3_O_4_ NPs	Fe_3_O_4_ NPs size < 10 nm	*S. aureus* *P. aeruginosa*	Excellent anti-adherence and anti-biofilm properties. Low toxicity and an easily biodegradable material	[156]
	Chitosan (1:5 Fe_3_O_4_ NPs/chitosan)	Chitosan- Fe_3_O_4_NPs composites Fe_3_O_4_ NPs coating Fe_3_O_4_ NPs form nd	Fe_3_O_4_ NPs size nd	*E. coli*	Antibacterial properties Dye absorbent	[157]
	Cathelicidin LL-37 (128 μg/mL) Ceragenin CSA-13 (0.5–8 μg/mL)	Fe_3_O_4_ NPsand peptides in solution Spherical Fe_3_O_4_ NPs	Fe_3_O_4_ NPs diameter ~12 nm	MRSA Xen 30, and *P. aeruginosa* Xen 5	Antibacterial properties	[158]

##### Silver Nanoparticles

Silver nanoparticles (AgNPs) are another promising nanomaterial to fight against infections caused by MDR bacteria. From ancient times, silver has been used due to its antimicrobial capacity. At present, silver is used in nanoparticles with applications in the food and medical industry [159,160,161,162]. Despite the enormous potential of these nanoparticles, some studies have shown that AgNPs have cytotoxic effects in different cell lines depending on their size, shape, concentration, or capping agent [163,164,165]. For this reason, combinations with other antimicrobial treatments have been proposed to improve the antibacterial activity and reduce the AgNPs’ cytotoxicity.

Many authors have shown that AgNPs combined with antibiotics synergize against MDR strains [166,167,168,169]. Li et al. synthesized AgNPs combined with antibiotics (vancomycin, oleandomycin, ceftazidime, penicillin G, novobiocin, carbenicillin, lincomycin, or erythromycin) showed a synergistic effect against MDR *P. aeruginosa* and *E. coli* [133]. In another study, AgNPs were conjugated with ampicillin and amikacin (1:1 ratio) and tested against clinical isolates of *E. faecium, S. aureus, A. baumannii, Enterobacter cloacae, E. coli, K. pneumoniae,* and *P. aeruginosa*. The results showed synergy, partial synergy, and additive effects among the conjugates [134].

Other studies have shown that the combination of bacteriocins and AgNPs is an effective treatment against ESKAPE pathogens [170,171,172,173]. For example, Gomaa used a bacteriocin extracted from *Lactobacillus paracasei* that, in combination with AgNPs, exhibited a more potent synergistic bactericidal effect than individual treatments against clinical MDR isolates of *S. aureus*, *P. aeruginosa*, *K. pneumoniae*, *E. coli**,* and *Staphylococcus pyogenes* [135].

One of the strategies to reduce the AgNPs cytotoxicity is using polymers as capping agents or for controlled release [174,175,176,177]. For example, Abdalla et al. used polyvinyl alcohol (PV) and chitosan (C) as capping agents in AgNPs and were tested against clinical isolates of *S. epidermis*, *S. aureus*, *K. pneumoniae*, and *E. coli*. Interestingly, this combination had a remarkable antimicrobial effect and inhibited biofilm production in all analyzed strains [136].

Furthermore, AgNPs can be produced by alternative forms. For example, Figueiredo et al. have carried out studies on fungal synthesis. In this study, the AgNPs were functionalized with simvastatin and then tested against reference and MDR bacterial strains. The combination showed antibacterial activity against MRSA strains and extended-spectrum beta-lactamase *E. coli*. These results demonstrated the great potential of this combinatorial treatment to combat bacterial infections [178]. Another example of an alternative synthesis of AgNP is photo-reduction. Courrol et al. functionalized AgNPs with tryptophan and evaluated their effect on antibiotic-resistant and susceptible pathogens. The results showed inhibition of ~100% (bactericidal effect) in *Salmonella thipymurium, P. aeruginosa, S. aureus,* MDR *S. epidermis,* MDR *K. pneumoniae*, and MDR *E. coli*. The inhibition of biofilm formation was also investigated, and data showed an inhibitory effect in *E. coli*, *K. pneumoniae*, *C. freundii*, *P. aeruginosa*, *S. aureus*, and *S. epidermidis* [179].

Finally, one of the most recent advances involving silver nanomaterials is silver phosphate nanoparticles (Ag_3_PO_4_ NPs). Among semiconductor nanomaterials, Ag_3_PO_4_ NPs have attracted considerable attention due to their photocatalytic activity, which has the potential to generate ROS for the degradation of organic dyes under visible light irradiation, making them a candidate for killing pathogenic bacteria [180,181,182]. Steckiewicz et al. evaluated the antimicrobial and antibiofilm properties of Ag_3_PO_4_ with different morphologies. The authors observed that the cubic morphologies had better antimicrobial activity, obtaining an 8 μg/mL MIC against *S. aureus* ATCC 25923 and MRSA ATCC 33591. In contrast, minimum biofilm eradication concentrations (MBECs) of 32 μg/mL and 16 μg/mL were observed for *S. aureus* ATCC 25923 and MRSA ATCC 33591, respectively. Unfortunately, these nanoparticles have cytotoxic effects on osteoblast line cells (hFOB1.19, MC3T3-E1, Saos-2, C2C12, and HD) starting at 5 μg/mL [183]. So far, there is limited literature on their antimicrobial properties and cytotoxicity. Therefore, more studies are needed to show the potential application of this promising nanomaterial.

##### Zinc Oxide Nanoparticles

Zinc oxide nanoparticles (ZnO NPs) are widely used in different areas, such as cosmetics, solar panel development, biosensors, and medicine [184,185,186]. Lately, ZnO NPs have received considerable attention due to their efficient bactericidal activity against multidrug-resistant bacteria [187,188]. Despite their excellent antimicrobial properties, some studies have shown the potential toxicity in different cell lines [189]. Some alternatives have been sought to counteract these effects, such as the combination with other antimicrobial compounds.

Several investigations have reported synergistic activity between ZnO-NPs and antibiotics. For example, Bhande et al. used ZnO NPs/β-lactam antibiotics-based combinatorial therapies to treat beta-lactamase-producing bacteria (clinical isolates from urinary tract infections). The combination of ZnO NPs with cefepime or ampicillin was synergistic in *E. coli*. In contrast, the combination of ZnO NPs with cefotaxime or ceftriaxone revealed a similar synergy pattern in *K. pneumoniae* [137]. In another study, the combination of ZnO NPs with ciprofloxacin and ceftazidime was tested against resistant *A. baumannii*. The results showed that the antimicrobial activity of ciprofloxacin and ceftazidime was increased in the presence of ZnO NPs. Thus, this combined strategy effectively overcame bacterial resistance [138]. Recently, Tyagi et al. synthesized ZnO-NPs conjugated with ciprofloxacin and evaluated its effect against *Klebsiella* spp. and *E. coli*. This conjugate increased the antibacterial activity against *Klebsiella* spp. and *E. coli* by 4.3- and 2.7-fold, respectively, compared to ZnO-NPs alone [139]. Moreover, a recent study investigated the effect of ZnO NPs in combination with meropenem, ciprofloxacin, and colistin against *P. aeruginosa* ATCC strain and clinical isolate strains of *P. aeruginosa*. Notably, the combination of ZnO NPs with colistin showed a synergistic effect, and it might be beneficial as a therapeutic alternative to *P. aeruginosa* infections [140].

During the last five years, different research groups have sought to use natural polymers in combination with ZnO NPs as a possible treatment against multidrug-resistant bacteria [141,190,191,192,193]. A group of researchers evaluated the antibacterial effects of NPs of chitosan, ZnO alone, and a combination of chitosan and ZnO against MDR and wild-type strains. ZnO combined with chitosan showed synergistic effectiveness on MDR *E. coli* and MDR *E. faecium* [141]. A few years later, Mehta et al. tested biofilm activity against MDR *E. faecium* with the previously described composite (ZnO-chitosan composite) in a lipid micelle. Interestingly, results showed a 50% reduction in bacterial biofilm size than chitosan and ZnO alone [142]. Finally, our research group has synthesized a novel biocomposite that serves as an antimicrobial against MDR *S. aureus* and MDR *P. aeruginosa*. This biocomposite was composed of an exopolysaccharide (EPS) from *Rhodotorula mucilaginosa* UANL-001L as a capping agent that previously showed antibacterial properties [144] and ZnO-NPs. We showed that the biocomposite inhibited bacterial growth up to 50 and 80% in MDR *P. aeruginosa* and MDR *S. aureus*. Furthermore, we have determined the toxicological effects of this biocomposite in a Wistar rat model. Our results showed that a three-day oral regimen of 6 mg/mL did not produce any toxic adverse effects in the rats, so we concluded that this composite has potential applications as an antimicrobial agent against resistant strains [143].

##### Titanium Dioxide Nanoparticles

Along with ZnO NPs, titanium dioxide nanoparticles (TiO_2_ NPs) are among the most promising metal oxide nanoparticles for pathogen control. Due to their whitish color, these NPs are used as an additive in the paint, cosmetics, and food industry [194]. Moreover, they have excellent catalytic properties and are widely used for water and air treatment, organic synthesis, fertilizer production, and, finally, as a product for decontamination and as a food preservative due to their antimicrobial activity [195,196]. Despite its wide applications, various studies have shown that TiO_2_ NPs are quite toxic in animal models and even classified as possibly carcinogenic in humans by The International Agency for Research on Cancer (IARC) thus reducing their harmful effects is vital to continue its application [195,197,198,199]. As seen throughout this review, combinatorial treatments can reduce toxicity by decreasing individual concentrations. Thus, combinatorial strategies seem to be an option to improve therapies with TiO_2_-NPs.

Many research groups have been dedicated to improving the antibacterial activity of TiO_2_ NPs by combining them with other nanoparticles or nanomaterials [200,201,202]. In the first example of TiO_2_ NPs-based combinatorial treatments, we present the study by Li et al., where they observed synergy between TiO_2_ NPs and two geometric isomers carborane derivatives (named FcSB1 and FcSB2) as a treatment against clinical isolates of MDR *A. baumannii*. Results showed that by combining fractional MIC values, up to 100% of MDR *A. baumannii* growth could be inhibited [145]. Then, Masoumi et al. studied the antimicrobial properties of TiO_2_ NPs, ZnO NPs, and synthetic peptides against MDR clinical isolates of *A. baumannii, K. pneumoniae,* and *P. aeruginosa*. This combination showed additive effects against *A. baumannii* and *K. pneumoniae*. The combination of nanoparticles (TiO_2_ NPs and ZnO NPs) had an additive effect on *A. baumannii* and *K. pneumoniae* strains [146]. Recently, Hérault et al. tested the antibacterial effect of silver-containing titanium dioxide nanocapsules against *E. coli*, *S. aureus*, and MDR *S. aureus*. The authors demonstrated that these nanocapsules had great antibacterial activity against wild-type and MDR *S. aureus*. Thus, this result showed the potential biomedical use of this therapeutic strategy [147].

Other studies have proven that the combination of TiO_2_ NP with different polymers can be used against MDR bacteria [203,204]. For example, Zhang et al. synthesized a nanocomposite of polytetrafluorethylene (PTFE) and TiO_2_-NPs against *E. coli* and *S. aureus*. In this study, the nanocomposite showed antibacterial and anti-adhesion properties against both bacteria, and it also demonstrated biocompatibility in fibroblast cells. Thus, this nanocomposite is a promising candidate for biomedical implants [148]. Recently, Harun et al. used a TiO_2_/ZnO nanocomposite against non-MDR and MDR bacterial strains (*S. aureus* ATCC29213, *E. coli* ATCC 25922, MRSA ATCC 38591, and *K. pneumoniae* ATCC 700603). Results showed the bactericidal activity of heterogeneous TiO_2_/ZnO (25T75Z molar ratio) nanocomposite in Gram-positive and -negative bacteria. Interestingly, the authors also observed a 50% reduction in the biofilm formation in MRSA and MDR *K. pneumoniae* [149].

Similarly, different research groups have used TiO_2_-NPs to improve the antibacterial activity of different antibiotics in MDR strains [205,206]. A recent study evaluated the antibacterial effect of TiO_2_-NPs in combination with antibiotics against MDR *P. aeruginosa* strains. The combination of TiO_2_-NPs with cefepime, ceftriaxone, amikacin, and ciprofloxacin exhibited synergistic activity against all tested isolates [150]. Finally, a recent study demonstrated the enhanced antibacterial activity of combination erythromycin/TiO_2_-NPs against MRSA. The combination was found to be more potent than individual treatments. This therapeutic strategy seems to be a potential alternative to conventional antibiotics to treat MRSA strains [151].

##### Magnetite Nanoparticles

Magnetite can be used as a nanomaterial due to its magnetic and biological properties, such as thermal properties and chemical and colloidal stability [207]. Although it is mentioned in some works with antibacterial and biofilm activities, the doses to achieve these effects are quite high [208]. Therefore, an alternative is to use it in combinatorial treatments.

One of the unconventional methods of sterilizing medical equipment is developing multifunctional bioactive coatings capable of inhibiting microbial proliferation. Within this method, we can find that structures formed by a spherical core of magnetite functionalized with cefepime (Fe_3_O_4_ @ CEF) and a thin layer of PLGA have been satisfactorily synthesized, which proved to be suitable materials for the sterilization of the surface of implantable devices, in terms of improved biocompatibility and efficient and constant inhibition of bacterial biofilm (staphylococcal and *Escherichia coli* colonization) [155]. Likewise, water-dispersible nanostructures based on magnetite (Fe_3_O_4_) and eugenol (E) have been synthesized. These nanostructures have proven to be another successful alternative to control and prevent infections associated with microbial biofilms of *S. aureus* and *P. aeruginosa*. This nanomaterial showed excellent anti-adherence and anti-biofilm properties, besides using bioactive natural compounds, representing lower toxicity and an easily biodegradable material [156]. Later, chitosan and magnetite nanoparticles biocomposites (Cs/Fe_3_O_4_) were synthesized for efficient dye adsorption and as an antibacterial agent. Besides the excellent adsorbent efficiency, the composite showed an antibacterial efficacy against *E. coli* [157]. Finally, a study reported the bactericidal activity of MNPs combined with the human antibacterial peptide cathelicidin LL-37, ceragenins, or classical antibiotics (vancomycin and colistin) against MRSA Xen 30 and *P. aeruginosa* Xen 5. The combination of LL-37 peptide or ceragenin CSA-13 with MNPs showed antibacterial properties, suggesting an alternative to developing new methods to treat infections caused by MDR bacteria [158].

Last, a research group conducted magnetite/silver/antibiotic (rifampicin, doxycycline, ceftriaxone, and cefotaxime) nanocomposites for targeted antimicrobial therapy. Antimicrobial properties of magnetite/silver/rifampicin and magnetite/silver/doxycycline nanocomposites were confirmed in *S. aureus* and *Bacillus pumilus* [152,154]. Then, the cytotoxicity of these nanocomposites was evaluated in human embryonic kidney 293 (HEK293T). Unfortunately, the results showed that magnetite/silver/rifampicin and magnetite/silver/doxycycline NPs induced cytotoxicity in the tested cell line [153]. Thus, more cytotoxicity studies should be performed before the clinical application of these nanocomposites.

#### 2.2.3. Antimicrobial Peptides

Antimicrobial peptides (AMPs) are a family of naturally occurring antimicrobial low-molecular-weight proteins with a broad spectrum of antimicrobial activity [209]. In recent years, research on antimicrobial peptides has increased. Several AMPs are in clinical development, and some others have undergone clinical trials; however, a few have been successfully commercialized [209,210,211]. AMPs include molecules such as defensins, cathelicidins, granulysin, S-100 proteins, and colistin, among others [212]. Colistin is one of the seven US Food and Drug Administration (FDA)-approved AMP [213]. The Gram-positive bacterium *Paenibacillus polymyxa* produces it and is currently being used as an antibiotic (last-resort drug) to treat multidrug-resistant Gram-negative bacteria [214,215].

##### AMPs-Based Combinatorial Treatments

Combinatory therapies mainly lead to synergism or additive antimicrobial effects; therefore, the combination has a stronger effect than a single drug. Moreover, the use of combinatory therapies represents a clinical improvement and significantly decreases the possibility of antibacterial resistance [216]. Several studies have shown the advantages of therapies based on AMPs and antibiotics against MDR and biofilm-forming bacteria [22]. A summary of the combinatorial treatments that include antimicrobial peptides as the main component of effective antimicrobial agents is presented in Table 3.

Human neutrophil peptide (HNP-1) is one of the most potent defensins produced by neutrophils [217]. A study showed that the combination of antituberculosis antibiotics isoniazid and rifampicin with HNP-1 had a better antimicrobial effect against *Mycobacterium tuberculosis* from infected lungs, liver, and spleen than they were employed alone [218]. The synthetic LL-37 is the only cathelicidin-derived antimicrobial peptide found in humans [219]. The evaluation of the antibacterial activity of amoxicillin with clavulanic acid and amikacin combined with the synthetic LL-37 peptide revealed a more significant killing effect against clinical isolates of *S. aureus* [220]. Another peptide is arenicin-1, which has been reported to exhibit broad-spectrum antimicrobial activity [221]. Combining this peptide with conventional antibiotics (ampicillin, erythromycin, and chloramphenicol) demonstrated synergistic activity and killed bacteria by interfering with the biosynthesis of DNS, proteins, or cell wall components [221].

A recent study has demonstrated that the synthetic cyclolipopeptide analog of polymyxin (AMP38) was tested in combination with carbapenems. The synergistic effect was observed to cause the killing of biofilm-forming and carbapenem-resistant *P. aeruginosa* [222]. On the other hand, the synergistic effects of antimicrobial peptide DP7 on some multidrug-resistant bacterial strains (*S*. *aureus*, *P. aeruginosa*, and *E. coli* were evaluated). The results showed that the combination of DP7 peptide with azithromycin or vancomycin was more effective, especially against highly antibiotic-resistant strains [223].

**Table 3 antibiotics-11-00794-t003:** Combinatorial treatments that include antimicrobial peptides as the main component of effective antimicrobial agents.

Nanomaterial	Combined with	Targeted Bacteria	Antimicrobial Effects	References
Human neutrophil peptide	Isoniazid and rifampicin	*Mycobacterium tuberculosis*	Antimicrobial effect	[218]
Synthetic LL-37 (cathelicidin-derived peptide)	Amoxicillin with clavulanic acid and amikacin	Clinical isolates of *S. aureus*	Significant killing effect	[220]
Arenicin-1	Ampicillin, erythromycin, and chloramphenicol	*Staphylococcus aureus* (ATCC 25923), *Enterococcus faecium* (ATCC, 19434), *Staphylococcus epidermidis* (KCTC 1917), *Pseudomonas aeruginosa* (ATCC 27853), *E. coli* (ATCC 25922), and *E. coli* O-157 (ATCC 43895)	Synergistic activity and kill bacteria by interfering with biosynthesis of DNS, proteins, or cell wall components	[224]
Synthetic cyclolipopeptide analog of polymyxin (AMP38)	Carbapenems	Carbapenem-resistant *P. aeruginosa*	Synergistic effect	[222]
Peptide DP7	Azithromycin or vancomycin	MDR strains (*S*. *aureus*, *P. aeruginosa*, and *E. coli*)	Antimicrobial effect	[223]
ASU014	Oxacillin	MRSA	Improved the killing effect	[225]
A broad set of AMPs	Ciprofloxacin, meropenem, erythromycin, and vancomycin	*Enterococcus faecium*, *S. aureus, K. pneumoniae*, *A. baumannii, P. aeruginosa*, *Enterobacter cloacae*	Synergistic effects	[226]

Recent studies have recently shown the effect of a broad set of AMPs combined with antibiotics against drug-resistant strains. Combinatorial treatment of ASU014, a bivalent branched peptide, with oxacillin, was very efficient against MRSA. The synergism between both meaningfully improved the killing effect compared to single drugs. Lower peptide concentrations and sub-MIC doses of the antibiotic were required for the complete eradication of the pathogen [225]. Another study evaluated the synergy between conventional antibiotics (ciprofloxacin, meropenem, erythromycin, and vancomycin) and a broad set of AMPs in a murine model induced by ESKAPE (*Enterococcus faecium*, *S. aureus*, *K. pneumoniae*, *A. baumannii*, *P. aeruginosa*, *Enterobacter cloacae*) pathogens. Combinatorial treatments showed synergistic effects that significantly reduced abscess sizes and/or improved clearance of bacterial isolates from the infection site, regardless of the antibiotic mode of action. Therefore, these results open new opportunities to develop alternative therapies [226].

As we mentioned, antimicrobial peptides are an up-and-coming alternative for the treatment of infections caused by multidrug-resistant bacteria; however, they present some disadvantages such as their reliability due to their susceptibility to proteases and changes in pH, short half-life, and rapid renal clearance [227]. Therefore, it is possible to use AMPs in combination with drug delivery systems, such as liposome encapsulation, inorganic (AuNPs or AgNPs), and polymeric (chitosan, or PLGA) nanoparticles [228].

All the previously described data strongly support the idea that combinatorial treatments have increased antibacterial activity. However, evaluating these treatments in animal models and clinical trials is necessary to assess their efficacy and safety. We have summarized in Table 4 some of the most relevant studies of nanomaterial-based combinational treatments where their antibacterial activity was evaluated using in vivo models.

Although promising results have been obtained in preclinical studies, additional clinical research is required to elucidate the efficacy of combination treatments for clinical practice. So far, a few clinical trials have ventured into the development of combinatorial therapies, including inorganic and polymeric nanoparticles and AMPs for antibacterial treatment. Clinical trials of nanomaterial-based combinational treatments are summarized in Table 5.

A phase III study is currently underway in the literature to test the efficacy of the new chitosan containing lesion sterilization therapy and tissue repair (LSTR) material in a dual antibiotic paste. This study has been published, but the registry is not yet open [231]. The incorporation of different nanomaterials has been proposed to improve the antimicrobial properties of bioceramic seals. For example, a phase IV study seeks to test the efficacy of a combination of AgNPs or chitosan with bioceramic sealants to improve its antimicrobial properties using an infection model. The results of this study are not published yet [232,233]. One of the basic requirements for a good treatment of caries is to find a way to seal these areas and isolate them from microorganisms so that they do not enter and colonize [234]. As an example, we have a phase III clinical trial that evaluates the antibacterial effect of a glass ionomer to seal caries when incorporated with chitosan and TiO_2_ NPs on carious dentin treated after partial removal of caries. Unfortunately, the results of this trial have not been published yet. On the other hand, a phase III clinical study evaluated the antibacterial effect of a new product called Nano Silver Fluoride ^®^ (a solution of AgNPs, chitosan, and fluoride) in occlusal carious molars treated with the partial caries removal technique to prevent and treat caries cavities. In this study, 44 people were brought together to analyze the antimicrobial properties of the so-called Nano Silver Fluoride ^®^. Of the 44 people, 22 were treated with Nano Silver Fluoride ^®^, and the rest with a control treatment. Surprisingly, Nano Silver Fluoride ^®^ manages to reduce the count of bacteria present in Molars Treated by up to 44%.

Regarding the AMPs, the results obtained in pre-clinical studies demonstrated their potential to be evaluated in clinical trials to assess their efficacy and safety. To date, 41 AMPs have been evaluated in clinical phases; some are still under study, and others have been completed, discontinued, or approved. Currently, five AMPs (nisin, gramicidin, polymyxins, daptomycin, and melittin) are in clinical use as an alternative to conventional treatments [235]. However, only a couple of studies have evaluated the combinatorial treatment as a strategy to protect or reduce the toxicity of AMPs. Nisin (nisin A) is naturally produced by lactic acid bacteria (*Lactococcus lactis*). Applications of nisin in humans include dental care and pharmaceutical products to treat stomach ulcers and colon infections. Nisin is degraded by enzymes in the gastrointestinal (GI) tract; therefore, it requires a delivery system to reach the site of action without being digested and absorbed during its passage through the GI tract. A strategy to improve this, nisin was encapsulated in pectin/ hydroxypropyl methylcellulose-coated tablets [236]. Polymyxins (A, B, C, D, and E) are a group of cyclic polypeptides which are used to treat eye (polymyxin B) and wound (polymyxin E) infections. These AMPs are last-resource treatment options because of their common side effects, including nephrotoxicity and neurotoxicity. To reduce the occurrence of adverse effects, polymyxin B has been administered as a pro-drug that, when hydrolyzed, produces the active AMPs. Moreover, polymyxin E has been successfully incorporated into hydrogels to treat burn wound infections [237].

**Table 5 antibiotics-11-00794-t005:** Clinical trials of nanomaterial-based combinational treatments.

Nanomaterial	Applied with	Trial Description	Clinical Trial Identifier
Chitosan nanoparticles	Double antibiotic paste	To evaluate the clinical double antibiotic, paste mixed with chitosan nanoparticles gel in lesion sterilization and tissue repair in non-vital primary molars.	NCT05079802
Silver nanoparticles and chitosan	Bioceramic sealer	To assess the antibacterial efficacy and adaptability of bioceramic sealer when incorporated with nanosilver.	NCT04481945
Titanium dioxide nanoparticles and chitosan	Glass ionomer	To study antibacterial effect on carious dentine of glass ionomer when modified with chitosan and titanium dioxide nanoparticles.	NCT04365270
Silver nanoparticles and chitosan	Fluoride	Evaluation of the antibacterial effect of nano silver fluoride on occlusal carious molars.	NCT03186261
Nisin	Pectin/hydroxypropyl methylcellulose-coated tablets	Delivery system to reach the site of action without being digested.	[236]
Polymyxin E	Hydrogels	To treat of burn wound infections.	[237]

## 3. Conclusions

Even though antibiotics have been a great weapon against pathogenic bacteria, it is urgent to create new therapies against pathogens that have developed strategies to overcome their mechanisms. Novel materials, such as polymers, nanoparticles, or AMPs, offer clear advantages over conventional antibiotics to combat various infectious diseases. However, it was also noted that some therapies have unwanted toxic effects on human cell lines. Only a few combinatorial treatments have reached the clinic. Challenges toward clinical applications of combination strategies include cytotoxicity effects, production costs, bioavailability, and efficacy. Several strategies have been designed to overcome these challenges, such as chemical modifications, delivery systems, and new synthesis technology (electrospinning and three-dimensional printing). The development of combinatorial treatments is a multidisciplinary field; recently, intense research on the synthesis strategies and sophisticated techniques to produce complex nanomaterials has led to the development of combinatorial therapies involving nanomaterials. Given the significant research in the field, it may be expected that humankind will greatly benefit from nanomaterials and their combinations in the very near future, especially in the treatment of multidrug-resistant bacteria.

## Figures and Tables

**Figure 1 antibiotics-11-00794-f001:**
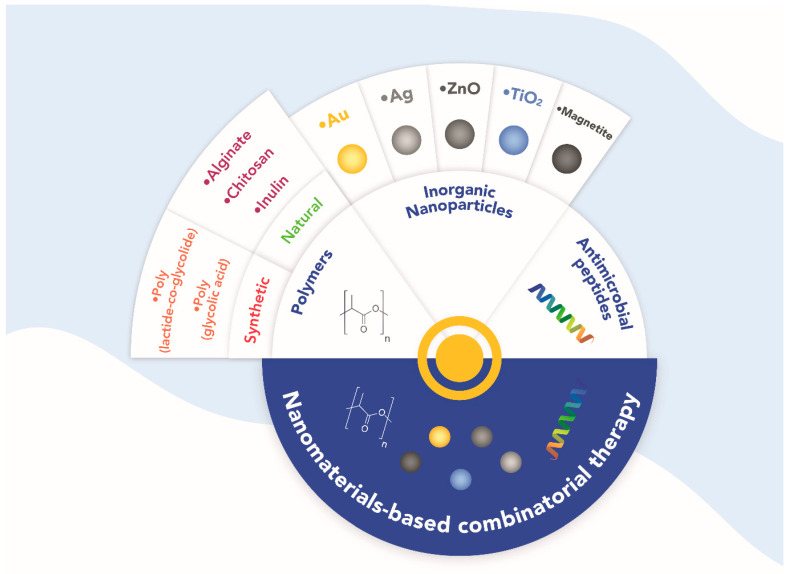
Combinatorial therapies used in the treatment of antibiotic-resistant bacteria.

**Figure 2 antibiotics-11-00794-f002:**
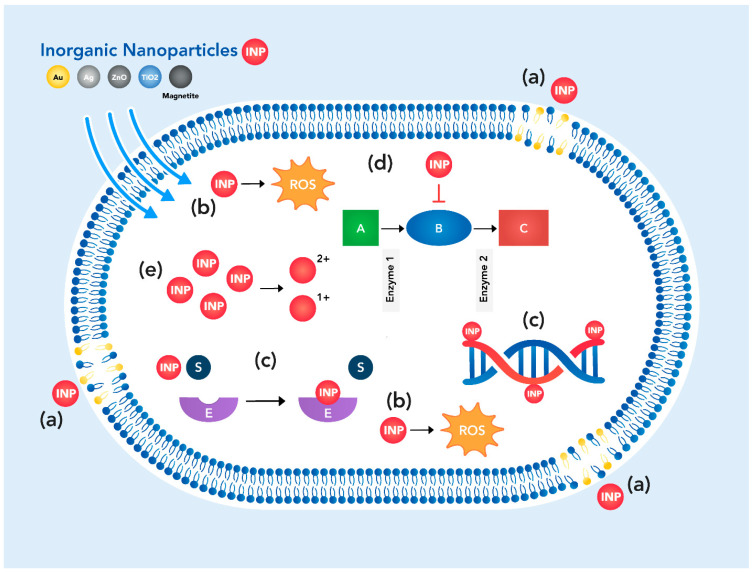
Schematic representation of antimicrobial mechanisms of inorganic nanoparticles (INP). (**a**) alteration of the bacterial cell wall and membrane, (**b**) induction of oxidative stress due to excessive intracellular production of reactive oxygen species (ROS), (**c**) inhibition of crucial proteins/enzymes and DNA, (**d**) disruption of metabolic pathways, and (**e**) intracellular accumulation of metal ions released from INP.

**Table 4 antibiotics-11-00794-t004:** Antibacterial activity of nanomaterial-based combinational treatments using in vivo models.

Nanomaterial	Combined with	In Vivo Model	Observations	References
Polymers				
Poly (lactide-co-glycolide)	Polycaprolactone Type I collagen AgNPs	Mouse periodontitis model	Novel silver-modified/collagen-coated PLGA/PCL scaffold features biocompatible, osteogenic, and antibacterial properties.	[45]
Poly (glycolic acid)	PLGA AuAg core/shell NPs	Farm pigs with stents implanted	The stent exhibited remarkable antibiofilm property and reduced the level of inflammatory and necrotic cells. The stent maintained structural integrity without the presence of large fragments in the urinary system	[51]
**Nanoparticles**				
Gold nanoparticles (AuNPs)	Bacterial cellulose (BC) 4,6-diamino- 2-pyrimidinotiol (DAPT)	Rat Wound Infection Model	The BC-Au-DAPT nanocomposites applied as wound dressings showed excellent anti-MDR bacteria activity and high biocompatibility.	[131]
Silver nanoparticles (AgNPs)	Bacteriocin (BC) extracted from *Lactobacillus paracasei*	*A. salina* model	BC/AgNPs bioconjugate was compatible to the biological system.	[135]
Zinc oxide nanoparticles (ZnO NPs)	EPS from *Rhodotorula mucilaginosa* UANL- 001L	Wistar rat renal model	EPS-capped ZnO NPs showed no toxic effect in vivo	[143,144]
Titanium dioxide nanoparticles (TiO_2_ NPs)	Silver ions	Female mice (Charles Rivers) macrophages and dendritic cells model	Despite uptake into macrophages, no proinflammatory response nor cytotoxicity in these cells were detected for our nanocapsules	[147]
**Antimicrobial peptides (AMPs)**				
PL-5	Levofloxacin	Mouse wound infection model	The synergistic application of PL-5 and levofloxacin inhibited bacteria, with a bacteriostatic rate of 99.9%	[229]
HNP-1	Silica nanoparticles	Rats wound infection model	Gels containing HNP-1 and showed a significantly faster wound healing in comparison with control.	[230]

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
