# Peer review of "Nanomaterials-Based Combinatorial Therapy as a Strategy to Combat Antibiotic Resistance"

_antibiotics, 2022, doi:10.3390/antibiotics11060794_

Round 1

Reviewer 1 Report

The Review entitled “Nanomaterials-based combinatorial therapy as a strategy to combat antibiotic resistance” is original and has a significance for the scientific community. Authors have analyzed 182 literature sources including the latest studies (2020-2022) along with covering of early publications (2000-2005). Review contains also authors’ own research publications that indicates their awareness of this issue.

Presented review is clearly structured and of great interest to a wide range of readers. Manuscript is focused mainly on the nanotherapeutic combinatorial treatments that target bacterial infections. The proposed for review item is relevant in context of combinatorial therapies application where nanomaterials have gained significant interest.

Bearing in mind the aforementioned, I highly recommend to accept this Review for publication.

Author Response

Jun 1, 2022

Response to Reviewer’s Comments

Reviewer #1:

Reviewer #1: The Review entitled “Nanomaterials-based combinatorial therapy as a strategy to combat antibiotic resistance” is original and has a significance for the scientific community. Authors have analyzed 182 literature sources including the latest studies (2020-2022) along with covering of early publications (2000-2005). Review contains also authors’ own research publications that indicates their awareness of this issue.

Presented review is clearly structured and of great interest to a wide range of readers. Manuscript is focused mainly on the nanotherapeutic combinatorial treatments that target bacterial infections. The proposed for review item is relevant in context of combinatorial therapies application where nanomaterials have gained significant interest.

Bearing in mind the aforementioned, I highly recommend to accept this Review for publication.

We greatly appreciate the reviewer for taking the time to evaluate the manuscript and for considering our work exciting regarding the idea of a review focused on combinatorial nanotherapies to treat infections caused by sensitive and resistant pathogenic bacterial strains. We are very excited about the feedback and for recommending it for publication in its current form.

Reviewer 2 Report

  1. Please improve introduction and conclusion.
  2. Authors should improve quality figures and also added mechanism figures for nanoparticles in the manuscript.
  3. Manuscript need table on its application in vivo animal model.
  4. The clinical applications and studies are missing.
  5. The English needs to be checked and corrected by a native English writer.
  6. References need to be past 5 years unless important.

Author Response

Jun 1, 2022

Response to Reviewer’s Comments

Reviewer #2:

Reviewer #2: Please improve introduction and conclusion.

We thank the reviewer for this observation, we have improved the introduction and conclusion sections of the revised manuscript.

Authors should improve quality figures and also added mechanism figures for nanoparticles in the manuscript.

We appreciate this comment made by the reviewer, we have now improved the quality of figures in the revised manuscript, and in addition, we have included a new figure (Figure 2) where the possible mechanisms explaining the inhibition of bacterial cell growth and/or cell death by nanoparticles are represented.

Manuscript need table on its application in vivo animal model.

We thank the reviewer for this relevant observation, We have now added Table 4. Antibacterial activity of nanomaterial-based combinational treatments using in vivo models, to the revised version of the manuscript.

The clinical applications and studies are missing.

We appreciate this observation made by the reviewer and consider it relevant, we have therefore added Table 5. Clinical applications and studies of nanomaterial-based combinational treatments; to the revised version of the manuscript.

The English needs to be checked and corrected by a native English writer.

We thank the reviewer for this observation, we have had native English speakers read over the manuscript and have made all of the corrections suggested, including fixing grammatical errors.

References need to be past 5 years unless important.

We thank the reviewer for pointing this out and would like to comment that we went through all of our references and although we found some of them are from more than 5 years ago, we consider them necessary to substantiate several of the concepts or studies presented in this review, so we decided to keep them.

Reviewer 3 Report

Here I present the review of the paper entitled “Nanomaterials-based combinatorial therapy as a strategy to combat antibiotic resistance” submitted to Antibiotics.

The article is narrative review on nanomaterial-based combinatorial therapies against drug resistant bacteria. Review covers new area which has not been recently summarized. Language quality is sufficient. Title is clear and informative. Authors cited 182 adequate references.

In general, this is well written article, however some changes needs to be done.

Issues

  • Authors should describe in more depth resistance patterns which are later mentioned in text and tables (eg. MRSA).
  • Authors should discuss factors influencing antimicrobial activity of nanomaterials (size, shape etc.)
  • Please describe antibacterial mechanism of action of described nanomaterials in more depth. Especially for inorganic nanoparticles where there is a lot of evidence.
  • Table 2 is cut.
  • In lines 242-243 authors wrote “Although the AuNPs may not 242 have antimicrobial activity by themselves”. I cannot agree with this sentence

See: https://doi.org/10.1016/j.matlet.2014.01.108, https://doi.org/10.1016/j.arabjc.2013.11.044

  • It was also proven that Ag3PO4 particles can have antibiofilm and antibacterial activity against S. aureus MRSA. Although this is not well-established Authors may include that in the review

See: https://doi.org/10.1155/2019/6740325

Kind regards,

Reviewer

Author Response

Jun 1, 2022

Response to Reviewer’s Comments

Reviewer #3:

Reviewer #3: Here I present the review of the paper entitled “Nanomaterials-based combinatorial therapy as a strategy to combat antibiotic resistance” submitted to Antibiotics.

The article is narrative review on nanomaterial-based combinatorial therapies against drug resistant bacteria. Review covers new area which has not been recently summarized. Language quality is sufficient. Title is clear and informative. Authors cited 182 adequate references.

In general, this is well written article, however some changes needs to be done.

We greatly appreciate the reviewer for taking the time to evaluate the manuscript and for considering our work exciting regarding the idea of a review focused on combinatorial nanotherapies to treat infections caused by sensitive and resistant pathogenic bacterial strains. We are very excited about the feedback and for recommending it for publication after the appropriate recommendations.

Issues

  • Authors should describe in more depth resistance patterns which are later mentioned in text and tables (eg. MRSA).

We thank the reviewer for this recommendation and completely agree with it. We have therefore, in the revised version of the manuscript, described more in depth the resistance patterns, included MRSA resistance.

  • Authors should discuss factors influencing antimicrobial activity of nanomaterials (size, shape etc.)

We thank the reviewer for this suggestion. In the revised version we added the size, shape, and form/ratio of the component (if available) in Tables 1 and 2. We consider it appropriate to include this information in tables instead of the text, in order to make the reading more fluid.

  • Please describe antibacterial mechanism of action of described nanomaterials in more depth. Especially for inorganic nanoparticles where there is a lot of evidence.

We appreciate this comment made by the reviewer and agree with it. Therefore, in the revised version of the manuscript, we have added a section (General mechanisms of antimicrobial action of nanomaterials) where we explain the possible mechanisms of inhibition of bacterial cell growth and/or cell death by nanomaterials. We also added Figure 2 where the mechanisms of action of inorganic nanoparticles are represented since there is more evidence.

  • Table 2 is cut.

We thank the reviewer for this observation, We have made the necessary editing changes in Table 2.

  • In lines 242-243 authors wrote “Although the AuNPs may not 242 have antimicrobial activity by themselves”. I cannot agree with this sentence See:https://doi.org/10.1016/j.matlet.2014.01.108, https://doi.org/10.1016/j.arabjc.2013.11.044

We appreciate this observation made by the reviewer and agree with the comment. We have now, in the revised version of the work added a brief description about the antimicrobial effect of AuNPs.

  • It was also proven that Ag3PO4 particles can have antibiofilm and antibacterial activity against S. aureus MRSA. Although this is not well-established Authors may include that in the review https://doi.org/10.1155/2019/6740325

We appreciate this observation made by the reviewer and agree with the comment. We have now, in the revised version of the work included, at the end of silver nanoparticles section, the research about Ag3PO4 particles where their antibiofilm and antibacterial activity against S. aureus MRSA has been demonstrated.

Reviewer 4 Report

In this manuscript, the authors provided a comprehensive review of Nanomaterials-based combinatorial therapy as a strategy to combat antibiotic resistance. The paper reviews recent advances in combinatorial therapy with nanotherapeutics such as polymers, inorganic nanoparticles, and antimicrobial peptides. This manuscript could be helpful for the development of various nanomaterials that can fight against bacterial infections. I would like to recommend its publication in this journal after addressing the following recommendations:

  1. The authors should include a section, revealing:

- the state of the art of the design of composite nanomaterials and combinatorial strategies, for example, inorganic NP coated with a polymer, polymer nanoparticles loaded with antimicrobial peptides, nanoparticles incorporated in polymers, etc.

- the synthesis strategies and techniques for the production of combinatorial nanomaterials such as electrospinning, 3D printing, etc.;

  1. The possible mechanisms explaining the inhibition of bacterial cell growth and/or cell death by combinatorial nanomaterial should be included in the text. The reason for the synergistic activity of these NPs should also be described.
  2. In Tables 1 and 2, the form and size of the combinatorial nanomaterial should be added. It is not clear which part is the outer, which is the inner, or in what form/ratio the components are mixed. Moreover, the reference should be given as a number, not in a firm such as “Góra et al., 2015”.
  3. The specific dimensions of the nanomaterials should be better revealed and described in more detail in the text;
  4. Please, highlight the input of your own research groups in the appropriate sections.
  5. Overall, a more critical view of the reviewed papers is needed. The disadvantages of each combinatorial nanomaterial could be better emphasized.
  6. The conclusion should include a perspective for the future development of these nanomaterials.

Author Response

Jun 1, 2022

Response to Reviewer’s Comments

Reviewer #4:

Reviewer #4: In this manuscript, the authors provided a comprehensive review of Nanomaterials-based combinatorial therapy as a strategy to combat antibiotic resistance. The paper reviews recent advances in combinatorial therapy with nanotherapeutics such as polymers, inorganic nanoparticles, and antimicrobial peptides. This manuscript could be helpful for the development of various nanomaterials that can fight against bacterial infections. I would like to recommend its publication in this journal after addressing the following recommendations:

We greatly appreciate the reviewer for taking the time to evaluate the manuscript and for considering our work exciting regarding the idea of a review focused on combinatorial nanotherapies to treat infections caused by sensitive and resistant pathogenic bacterial strains. We are very excited about the feedback and for recommending it for publication after the appropriate recommendations.

  1. The authors should include a section, revealing:

The state of the art of the design of composite nanomaterials and combinatorial strategies, for example, inorganic NP coated with a polymer, polymer nanoparticles loaded with antimicrobial peptides, nanoparticles incorporated in polymers, etc.

We appreciate this comment made by the reviewer and agree with it. Therefore, we have added information about the design of composite nanomaterials and combinatorial strategies using INPs, polymers and AMPs into appropriate sections throughout the revised manuscript.

The synthesis strategies and techniques for the production of combinatorial nanomaterials such as electrospinning, 3D printing, etc.;

We appreciate this comment made by the reviewer and agree with it. Therefore, we have added a section focusing on electrospinning and 3D printing as synthesis strategies and techniques to produce nanomaterials.

The possible mechanisms explaining the inhibition of bacterial cell growth and/or cell death by combinatorial nanomaterial should be included in the text. The reason for the synergistic activity of these NPs should also be described.

We appreciate this comment made by the reviewer and agree with it. Therefore, in the revised version of the manuscript, we have added a section (General mechanisms of antimicrobial action of nanomaterials) where we explain the possible mechanisms of inhibition of bacterial cell growth and/or cell death by nanomaterials. We also added Figure 2 where the mechanisms of action of inorganic nanoparticles are represented since there is more evidence.

In Tables 1 and 2, the form and size of the combinatorial nanomaterial should be added. It is not clear which part is the outer, which is the inner, or in what form/ratio the components are mixed. Moreover, the reference should be given as a number, not in a firm such as “Góra et al., 2015”.

We thank the reviewer for this suggestion. In the revised version we have added to Tables 1 and 2, the form and size of the nanomaterials, including form/ratio of the component, if available. Regarding the reference issue, we have fixed it in the revision of the manuscript.

The specific dimensions of the nanomaterials should be better revealed and described in more detail in the text;

We thank the reviewer for this suggestion. In the revised version we added the size, shape, and form/ratio of the component (if available) in Tables 1 and 2. We consider it appropriate to include this information in tables instead of the text, in order to make the reading more fluid.

Please, highlight the input of your own research groups in the appropriate sections.

A great suggestion made by the reviewer, within the revised version of our work we have now highlighted the input of our own research in the appropriate sections.

Overall, a more critical view of the reviewed papers is needed. The disadvantages of each combinatorial nanomaterial could be better emphasized.

A great suggestion made by the reviewer, within the revised version of our work we have now emphasized the disadvantages of each combinatorial nanomaterial into the appropriated section.

The conclusion should include a perspective for the future development of these nanomaterials.

A great suggestion made by the reviewer, within the revised version of our work we have now improved the conclusion section appropriately.

Round 2

Reviewer 3 Report

In my opinion, the authors provided necessary changes.  Well done!

In my version of the manuscript there were only captions, but without figures, thus I cannot asses them. 

Kind regards

Reviewer 4 Report

The manuscript has been substantially improved and in its present form it may be suitable for publication.